# Working during a Pandemic between the Risk of Being Infected and/or the Risks Related to Social Distancing: First Validation of the SAPH@W Questionnaire

**DOI:** 10.3390/ijerph18115986

**Published:** 2021-06-02

**Authors:** Daniela Converso, Andreina Bruno, Vincenza Capone, Lara Colombo, Alessandra Falco, Teresa Galanti, Damiano Girardi, Gloria Guidetti, Sara Viotti, Barbara Loera

**Affiliations:** 1Department of Psychology, University of Turin, 10124 Turin, Italy; daniela.converso@unito.it (D.C.); lara.colombo@unito.it (L.C.); sara.viotti@unito.it (S.V.); 2Department of Education Sciences, University of Genoa, 16128 Genova, Italy; andreina.bruno@unige.it; 3Department of Humanities, University of Naples Federico II, 80133 Naples, Italy; vincenza.capone@unina.it; 4FISPPA Section of Applied Psychology, University of Padua, 35131 Padua, Italy; alessandra.falco@unipd.it (A.F.); damiano.girardi@unipd.it (D.G.); 5Department of Psychological, Health and Territory Sciences G. d’Annunzio, University of Chieti-Pescara, 66100 Chieti, Italy; teresa.galanti@unich.it (T.G.); gloria.guidetti@unich.it (G.G.)

**Keywords:** occupational health, COVID-19, perceived safety assessment

## Abstract

Background: The COVID-19 pandemic led the worldwide healthcare system to a severe crisis in which personnel paid the major costs. Many studies were promptly dedicated to the physical and psychological consequences of the COVID-19 exposure among healthcare employees, whereas the research on the other working populations has been substantially ignored. To bridge the current lack of knowledge about safe behaviors related to the risk of COVID-19 contagion at work, the aim of the study was to validate a new tool, the SAPH@W (Safety at Work), to assess workers’ perceptions of safety. Methods: A total of 1085 participants, employed in several organizations sited across areas with different levels of risk of contagion, completed an online questionnaire. To test the SAPH@W validity and measurement invariance, the research sample was randomly divided in two. Results: In the first sub-sample, Confirmatory Factor Analysis demonstrated the adequacy of the SAPH@W factorial structure. In the second sub-sample, multi-group Confirmatory Factor Analysis revealed that the SAPH@W was invariant across gender, ecological risk level, and type of occupation (in-person vs. remote working). Conclusions: The study evidenced the psychometric properties of the SAPH@W, a brief tool to monitor workers’ experiences and safety perceptions regarding the COVID-19 risk in any organisational setting.

## 1. Introduction

In organisational contexts, the ongoing COVID-19 emergency has brought about radical changes—in a noticeably short period of time—in terms of work processes and professional and social relations without the possibility of supporting or accompanying them [1]. The literature has highlighted the impact on psychological well-being related to the general fear of contagion [2,3], isolation during the lockdown period(s), concerns for one’s own health and the health of loved ones, and/or the loss of work and income [4,5]. Moreover, for the entire working population, an additional risk of developing stress and psychological discomfort arises from the risk (real and perceived) of being infected and/or—depending on whether the condition is experienced “in person” or “remotely” —the negative consequences of being “socially distanced” (e.g., technostress, the impossibility of separating family and workspaces and times, difficulties in coordinating work, etc.) [6,7]. All these risks may then generate additional emotional pressure and reduce safety perceptions, increase work injuries, and even encourage counter-productive behaviours [8].

While “essential workers” external to the healthcare sector (which is not considered in this study due to its specificity in view of biological hazards and infection risks) [9] have experienced since March 2020 the need to wear personal protective equipment (PPE) and to adopt safe behaviours in order to reduce the risk of contagion, a wider percentage of workers only experienced this new condition at a later stage. At the end of the first lockdown period (for Italy and most European countries, this occurred in early June 2020), several initiatives were then developed (see the dedicated websites of the main international Occupational Safety and Health Associations as OSHA, NIOSH; and ILO) [10,11,12] to facilitate a “safe return to work in phase 2” both for employees who had worked from home and for workers in production sectors that had been forced to shut down by the pandemic [13].

Phase 2 was not, as has been ascertained worldwide since autumn 2020, the end point, but just one of several phases of our indefinite period of coexisting with COVID-19. The need to ensure appropriate occupational health for all workers affected directly and indirectly by this pandemic and to increase knowledge and insights in this area [14], while maintaining the perception of the contagion risk and of organisational safety, controlling our own safe behaviours, has thus become even more important and has been highlighted by occupational health scholars and professionals [15,16].

Since 2020, many studies were promptly dedicated to the physical and psychological consequences of the COVID-19 exposure. We identified two main gaps in the literature. The first concerns the contagion risk perception, along with safe behaviours among workers external to the healthcare sector during the COVID-19 pandemic. Since February 2020, several scholars have, in fact, analysed the issue of perceived safety, fear of contagion, and the use of PPE only among medical and nursing staff [17,18,19,20,21]. The second gap, regardless of the pandemic period, concerns the fact that the literature on organisational safety largely focuses on the bio-risk hazard from a technical point of view. This type of contribution does not cover the need to comprehend workers’ perceptions of safety, as the development of a “complete” safety condition is related, on one hand, to the reliability of the technical safety measures adopted, and, on the other, to the condition of psychological safety which, to a large extent, although not exclusively, is based on that reliability.

To bridge these gaps, we developed a short self-report tool to assess workers’ perceptions of safety in organisations focused on the specific risk of COVID-19 contagion, also considering Non-Technical Skills (NTS) domain. As Flin and colleagues [22] have highlighted, a key determinant of workplace safety is “to focus on the non-technical skills of the system operators based at the ‘sharp end’ of the organisation. These skills are the cognitive and social skills required for efficient and safe operations”. They sustain safety performance in multiple and differentiated ways [23] and across different working sectors (aviation, healthcare, manufacturing organisations) [24]. In addition to the focus on workers’ NTS, our proposal was specifically focused on the organisational barriers and facilitators that promote or, on the contrary, may inhibit the use of the NTS required to protect safe and healthy work practices. For example, workers who have access to a wealth of safety information may perceive their risk level as being low, while unshared safety information may lead to lower levels of safety efficacy and higher levels of risk perception [25]. In this way, workers’ perceptions of safety are dependent upon the organisational conditions that are enacted by managerial choices and actions [26,27]. In the current COVID-19 epidemic, workers who have never before faced (such as, for example, healthcare sector operators) a biological risk or received specific training are attentive to, and evaluate, albeit not explicitly, safety at work by assessing not only their own performances, but the organisational sensitivity to the use of NTS for protecting a healthy and safe work environment.

This was the background to the development of a tool aimed at monitoring the perceived safety related to COVID-19 contagion and at highlighting those specific organisational skills required to consolidate the feeling of safety in the workplace for as long as COVID-19 presents a risk [13]. The objective of this study was to validate the new tool, namely SAPH@W (Safety at Work), using confirmatory factor analysis (CFA) and multiple-group CFA.

## 2. Materials and Methods

The studies presented in this article conform to the provisions of the Declaration of Helsinki (1995), revised in Edinburgh 2000. All ethical guidelines were followed in adherence to the legal requirements of the EU GDPR 2016/679. The front page of the online questionnaire clearly explained the research aims, the voluntary nature of participation, the anonymity of the data, and the elaboration of the findings, and requested the participants’ informed consent.

### 2.1. Procedure

Data were collected by means of the online administration of a standardised questionnaire. Participants were recruited using a snowball procedure, in different regions, to ensure that they originated from the majority of the Italian territory.

The combination of period and location of interviews represented a sufficiently heterogeneous sample of contagion risk levels, as recorded using the Rt index (i.e., the effective reproduction number, an indicator of the disease spread level) published by the Italian Ministry of Health and the Italian National Institute of Health during the COVID-19 pandemic.

To achieve the aim of validating the SAPH@W, the research sample was randomly divided into two: the first sample was used to test the item reliability and factorial structure of the SAPH@W, while the second sample was used to determine its measurement invariance across groups defined by gender, ecological risk level (low or high on the basis of the Rt index: below or above the value of 1) and type of occupation (in person vs. remote working-full or partial). 

### 2.2. Participants

There were initially 1089 respondents, but 4 of them ignored more than 10% of the items and were excluded from the database of answers. The research sample was thus ultimately composed of 1085 participants, working in public or private organisations, with the exclusion of the healthcare sector. Specifically, the occupational composition of the sample was: 43.6% administrative officers/employees, 21% factory/generic workers, 19.1% academic teachers, 11% qualified technicians/practitioners, 3.7% salespersons, and 1.6% executive managers/CEO. About one-third of participants lived and was employed in the north of the country (30%), while the rest resided and worked in central (34.7%) or southern (35.3%) regions. The mean age was 44.2 years (SD = 12.8). Among participants, 51.7% were female and 60.5% were married or in a stable cohabiting relationship; 50.6% of the sample were parents and 28.7% of them were responsible for a child aged 0–14; 11.6% of participants took care of (and lived with) a person having a certified disability or suffering from a chronic illness. In the sample, workers had been in their jobs for periods ranging from 1 to 50 years, with a mean of 14.7 years (SD = 11.8), and only 13.1% of the interviewed citizens declared the use of public transport to travel to work.

All working citizens involved voluntarily agreed to participate in the data collection procedure during the year 2020 (May–November).

### 2.3. Measures

The questionnaire included socio-demographic aspects, the SAPH@W scale, five items on perceived organisational adequacy during the COVID-19 pandemic, and a few questions on working conditions (remote or not, partly remote work or fully remote work, interacting or not with users/customers).

The participants’ evaluation of organisational adequacy was structured into five Likert type items concerning the organisational ability to rearrange practices, habits, spaces, shifts, and rules to cope with the COVID-19 pandemic and to guarantee the safety of employees.

SAPH@W was developed in close analogy with the NTSC-Q questionnaire, a tool for the self-assessment of non-technical skills created for chemical plant workers by Mariani and colleagues [28], due to its characteristics of covering the main domains of Flin’s taxonomy [22], while being brief and validated in Italian. The wording was modified from a focus on the worker to a focus on the organisation and on the current pandemic. In addition, a further scale was added, to assess the respondent’s perception of her/his contribution to healthy and safe work practices, in relation to each of the four dimensions involved: situational awareness, decision-making, communication, and fatigue management.

Globally, the SAPH@W consisted of 20 items developed to explore 5 content domains of the perception of safety at work during the COVID-19 pandemic; the majority of items regarded the perception of safety among employees deriving from the organisational reaction to the pandemic, while a set of items was devoted to measuring the personal contribution of workers in managing the COVID risk at work, i.e., how they feel able to adopt functional behaviours to facilitate and improve all measures to combat contagion. In detail, using a 5 points answers scale (from 1 “Not at all” to 5 “Completely”) the SAPH@W asked participants to evaluate the:Efficacy and effectiveness of the organisational communication on COVID-19 Promptness, foresight and care of the organisational decision-making process regarding COVID-19.Situational awareness of contagion risks and trends in the workplace.Organisational ability to recognise and care for workers’ fatigue specifically due to the pandemic (e.g., use of prevention devices, isolation, cognitive and emotive burden).Personal contribution to workplace safety in relation to COVID-19 (with regard to the four dimensions cited above).

The items are presented in the following Table 1.

### 2.4. Data Analysis

The research sample was randomly divided into two, ensuring that the two sub-samples (namely Sample A and Sample B) maintained the same composition in terms of risk levels and occupation type: the first sample was used to test the construct validity of the SAPH@W using confirmatory factor analysis (CFA)**,** while the second sample was used to determine its measurement invariance across groups defined by gender, ecological risk level (low-high), and occupation type (in person vs. remote working) using multiple-group CFA.

The description of the samples is presented in Table 2.

Confirmatory factor analyses (CFAs) were carried out using the maximum likelihood method with robust standard errors and a scaled test statistic (i.e., robust maximum likelihood, MLR) as estimator [29,30]; the full information maximum likelihood (FIML) was used to handle missing data [31]. To evaluate model fit, the scaled chi-square test was used together with additional fit indices, namely the root mean square error of approximation (RMSEA), the comparative fit index (CFI), and the standardised root mean square residual (SRMR). A model shows a good fit to data if the chi-square is non-significant. Values close to or smaller than 0.08 for RMSEA and SRMR and values close to or greater than 0.90 for CFI indicate acceptable model fit, whereas values close to 0.06 and 0.95 for RMSEA and CFI, respectively, indicate good fit [32]. Furthermore, the Akaike Information Criterion (AIC) was used to compare non-nested models. The model with the smallest AIC value is deemed to fit the data best among the competing models. For each latent factor of the SAPH@W, the average variance extracted (AVE) was calculated [33]. An AVE greater than 0.50 indicates that the latent factor accounts for most of the variance in its observed indicators, on average [34]. All the CFAs were estimated using the lavaan package version 0.6–7 [17] for R software version 4.0.3 [35].

To examine measurement invariance, multi-group CFAs were conducted, involving a sequential model testing approach [36]. Starting from configural invariance, more constraints on different sets of parameters across groups were progressively imposed, so as to test metric invariance [37] and scalar invariance [38]. In the presence of a significant test on the difference between the χ_2_ of the subgroups model, relatively invariant fit indices were considered indicative of an invariant factorial structure; a ΔCFI of 0.002 was considered appropriate invariance tests [36,39].

Finally, the differences in latent means of groups were analysed [40].

## 3. Results

### 3.1. Descriptive Analysis

An initial descriptive analysis was performed. Table 3 shows the mean (*M*), standard deviation (SD), asymmetry, kurtosis, and minimum and maximum value of all items, as resulting from the two sub-samples. The results are similar in the two sub-samples: the mean values of the items are above the average of the response scale (i.e., 2.5); the items related to the *fatigue management* dimension present the lowest mean values compared to the other dimensions. These results are in line with the asymmetry of the items, for both sub-samples: the asymmetry of all items is negative, except for the items of the *fatigue management* scale which present positive asymmetry. With regard to the kurtosis, the K values of items are <0 for both sub-samples, highlighting a platykurtic form of data distribution, with the exception of the item labeled COM2.

The internal homogeneity of the overall scale, calculated separately on Sample A and Sample B, is particularly good. Cronbach’s alpha on the overall scale is 0.95 in both samples and the same coefficient was found to be very good when calculated for each sub-scale, with values higher than 0.87.

### 3.2. Confirmatory Factor Analysis on Sample-A

To test the validity of the hypothesised 5-factor structure, a confirmatory analysis was performed on Sample A. Model fit was appreciable (χ^2^ (160) = 508.445 (*p* < 0.000), CFI = 0.955, RMSEA = 0.063, 90% CI = 0.057–0.069, SRMR = 0.038, AIC = 22,442.886), even though the solution clearly revealed the need to add a covariance between the error terms of item 3 and 4 of the *fatigue management* dimension, which has a very similar wording. When adding this parameter, the model fit increased: χ^2^ (159) = 365.778 (*p* < 0.000), CFI = 0.973, RMSEA = 0.049 (90% CI= 0.042–0.055), SRMR = 0.034, AIC = 22,254.616.

The correlations between the latent factors measured by the SAPH@W ranged from 0.434 (fatigue with personal contribution) to 0.811 (decision with awareness).

The high correlation between Decision-Making and Situational Awareness (0.811) suggested that a solution in which items from these two sub-dimensions loaded on a single latent factor might adequately fit the data as well. From this perspective, a 4-factor structure was also tested. However, compared with the 5-factor structure, the model fit decreased and was not satisfactory: χ^2^ (164) = 940.877 (*p* < 0.000), CFI = 0.900, RMSEA = 0.093 (90% CI = 0.088–0.099), SRMR = 0.045, AIC = 23,003.843, BIC = 23,287.574. This led to the alternative 4-factor solution being excluded.

The SAPH@W measurement model imposed on Sample A which was found to be more adequate is presented in Figure 1, where the factor names are abbreviated (com = *communication*, decmak = *decision-making*, sitaw = *situational awareness,* fatgman *= fatigue management,* persc *= personal contribution).*

### 3.3. Confirmatory Factor Analysis on Sample-B

To test the robustness of the SAPH@W structure that emerged in Sample A, the same model (i.e., five correlated factors and no cross-loading) was imposed on Sample B (N = 543). Although the model fit was acceptable (χ^2^ (160) = 395.28 (*p* < 0.000), CFI = 0.967, RMSEA = 0.052, 90% CI = 0.046–0.058, SRMR = 0.038, AIC = 22,861.020), the modification indices revealed that specifying item residual covariances between items 3 (“*Adopt measures to reduce physical fatigue*”) and 4 (“*Adopt measures to reduce mental fatigue*”) of the *fatigue management* dimension would considerably improve model fit, which had already occurred in Sample A. In fact, the modified model produced the following fit indices: χ^2^(159) = 302.21 (*p* < 0.000), CFI = 0.980, RMSEA = 0.041 (90% CI: 0.035–0.047), SRMR = 0.035, AIC = 22,736.832.

To verify that the items designed to measure each dimension functioned as expected, factor loadings were analysed (Table 4). They were all greater than 0.60 (0.68–0.95), and the Average Variance Extracted (AVE) indices were greater than 0.50 (0.64–0.86). Moreover, with regard to the association between the dimensions, the results revealed that the correlation indices between factors ranged between 0.30 (between *Fatigue management* and *Personal contribution*) and 0.83 (between *Decision-making* and *Situational awareness*).

### 3.4. Measurement Invariance

#### 3.4.1. Invariance across Gender

The model was estimated separately for male and female (Male, *n* = 283, reference; Female, *n* = 260). Independent CFA models demonstrated a good fit to the data, indicating that a multiple-group CFA was appropriate. Multi-group CFAs were conducted to examine Configural, Metric, and Scalar Invariance. The overall and comparative fit statistics of invariance models are presented in Table 5. The test of configural and metric (factor loadings were constrained to be equal across gender groups) invariances revealed good levels of fit. Inspection of the invariance test and Δχ^2^ did not show any significant differences between the baseline model and the metric model, and the ΔCFI was lower than 0.01. Standardised factor loadings of items were all significant. In metric invariance, factor loading ranged from 0.65 to 0.94 in males and from 0.71 to 0.96 in females. The scalar invariance model demonstrated good fit statistics: the CFI change between the two nested models was lower than the recommended criterion (ΔCFI = 0.001), indicating that full scalar invariance for gender was met.

Finally, the comparison between latent means highlighted that female workers reported lower scores in *Situational awareness* (Δ mean = −0.17, *p* < 0.05) and *Personal contribution* (Δ mean = −0.19, *p* < 0.01). All the latent mean differences for this and the following multi-group analyses are reported in the Appendix A.

#### 3.4.2. Invariance for Risk Area in Which the Participant Worked

The participants were classified, according to the risk level of the area in which they worked, in the “low-risk zone” (*n* = 298, reference) and “high-risk zone” (*n* = 245). The classification was made considering the regional Rt, as a description of the epidemic situation (data from the “Italian National Institute of Health”, https://www.iss.it, accessed on 22 January 2021), in the period of administration of the questionnaire. Independent CFA models, specified by risk zone of work, demonstrated a good fit to the data, indicating that a multiple-group CFA was appropriate.

Configural, Metric, and Scalar Invariance were implemented (Table 6). The test of configural and metric invariances revealed good levels of fit. Inspection of the invariance test and Δχ^2^ showed some significant differences between the baseline model and the metric model, but the ΔCFI was lower than 0.002. In metric invariance, factor loadings ranged from 0.68 to 0.95 in the “low-risk zone” participants and from 0.67 to 0.97 in the “high-risk zone” participants and they were all significant. The finding suggested that the model fit did not change substantially after constraining the factor loadings to be equal across groups. In addition, in the scalar invariance, the fit indices were in an appropriate range (ΔCFI < 0.002), indicating that full scalar invariance for the risk zone level was gathered.

The comparison between latent means underlined as “high-risk zone” workers scored lower in *fatigue management* (Δ mean = −0.25, *p* < 0.01).

#### 3.4.3. Invariance for Employment Status: In Person vs. Remote Working

The participants were classified, according to their employment status, as “in person” (*n* = 260, reference) and “remote” workers (*n* = 283). All parameters estimated were statistically significant for both groups (M1 and M2). The test of invariance and presentation of parameter estimates for employment status is reported in Table 7. The test of the model fit for the baseline model (M3) retained from the CFA, indicated a good model fit. The results showed that configural invariance was met.

In both the metric and scalar invariance models (M4 and M6), despite the fit indices being in an acceptable range, inspection of the invariance test, Δχ^2^ and ΔCFI (|ΔCFI| > 0.002), showed some significant differences between the respective baseline model and the others. Item 2 of dimension “decision-making” (“*When required, make quick decisions”*) had loading not invariant. Therefore, equality constraints were removed for this item, since the modification indices suggested that the constraint exclusion significantly improved model fit. We explored partial metric invariance (M5), and this final model presented an acceptable fit.

Lastly, a scalar invariance was tested. The results were not supportive of the scalar invariance, indicating that at least some intercepts were not equal across the samples. As suggested by modification indices, the intercept was freed for item 1 (“*Recognise the possible effects on physical fatigue”*) of dimension *fatigue management***.** The intercept of this item was higher for in person workers (*τx* = 2.85 vs. 2.69). Partial scalar invariance model M7 showed both good fit and an insignificant increase in model misfit compared to the partial metric (M5) model, indicating partial scalar measurement invariance. In relation to the latent means (κ), remote workers presented lower scores in communication (Δ mean = −0.21, *p* < 0.01), decision making (Δ mean = −0.18, *p* < 0.05), and situational awareness (Δ mean = −0.19, *p* < 0.05).

## 4. Discussion

The SAPH@W scale has demonstrated good psychometric properties, which remain stable in the two samples examined. In both samples, the responses are structured according to the theoretical expectations that guided the formulation of the SAPH@W scale. The perception of safety of workers can be expressed in four dimensions that qualify the organisation’s capacity to manage the return to work during the COVID-19 pandemic, namely situational awareness, capacity to communicate and make decisions effectively and efficiently, and the capacity to recognise additional mental and physical fatigue, generated by the pressure of working in the presence of an “invisible enemy” and by the stress of having to use devices and measures necessary to contain the contagion risk. In addition, as hypothesised, the perception of working safely includes the assessment of the personal contribution given by the workers themselves, namely the perception of effectiveness with respect to the adoption of behaviours functional to containing the risk.

The CFA carried out on both samples demonstrates that the scale is structured in 5 factors, well-specified, with no item in co-domain. The results also highlight that two items of the *fatigue management* dimension are closely linked, and thus cause covariance between the respective errors of measurement. However, this part of residual variability of the responses that unites the two items clearly depends upon the wording used, which is essentially the same, except for the subject of the statement: physical rather than mental fatigue. In fact, this specific aspect is not present in the remaining items of the dimension, which refer to evaluations of the states and causes of fatigue.

The SAPH@W scale has demonstrated that it is equally valid for men and women: the functioning of the scale is the same in the two gender sub-groups, with no difference in terms of loadings or intercepts in the items. With respect to gender, the SAPH@W scale is full invariant. Differences emerge, on the other hand, in the average values relating to the situational awareness and personal contribution dimensions, which were significantly lower in the sub-group of women. It is possible that this result may be linked to an actual gender difference in the severity of judgments relating to awareness and in the greater personal caution when faced with a risk, or the tendency to be more diligent, attentive, and respectful of security measures by women. In other words, the result may be explained by the lower propensity to risk of women, which leads them to be more scrupulous in adopting prevention measures and more diligent, and therefore severe, in assessing the organisation’s capacity to deal with the changing work conditions [41,42]. With respect to the different assessment of one’s own personal contribution, it is possible that this depends upon the greater inclination of women to underestimate themselves, considering themselves less effective and capable, as ascertained several times in many different research contexts. From school, in fact, despite achieving a better performance, girls tend to underestimate their own abilities.

This difference in average values may also depend upon the different occupations and duties performed by women, who, more frequently than men, perform jobs in the field of care, relationships, or contact with the public.

The analyses by sub-groups of workers allowed us to highlight that the functioning of the SAPH@W is full invariant also for risk levels: the structure and parameters estimated by the model are the same in the two sub-groups of workers employed in areas at high and low risk of contagion. The only difference between the groups is substantive and concerns the averages of the factor of recognition of physical and mental fatigue by the organisation: the scale is able to discriminate between those who, working in high risk areas, do not feel sufficiently recognised by the organisation with respect to the addition of burdens, emotional and physical, caused by the new working conditions. Strategies to mitigate the effects of fatigue in occupational settings are primarily centered on regulatory and organisational approaches, including limits on hours of service and efforts to ensure recovery from work. Such approaches are sensible and can generally have a positive effect on workplace alertness, safety, and productivity [43]. These strategies, in particularly risky situations, are not sufficient to guarantee the management of employee fatigue. The systematic use of prevention measures and the daily experience of a high probability of contagion could generate in workers the expectation of specific recognition, which does not seem to have been satisfied by their companies.

With respect to the working condition, in person or remotely, the analysis by sub-groups highlighted a partial invariance; two items violate the invariance: the first in terms of loading, and the second as intercept. Working in person rather than remotely accentuates the salience that workers attribute to the organisation’s capacity to make decisions rapidly and increases the severity of the judgment regarding the organisation’s capacity to recognise the physical fatigue of the worker during the pandemic. Despite the two violations, the measurement of the SAPH@W remains valid and discriminating, making a different position of the two groups of workers on factors relating to communication and situational awareness. For the group working in person, compared to the one working remotely, the organisation was more effective in understanding promptly the evolution of the risk in the workplace, as well as being more capable of developing prompt, effectively accessible, and useful communication. Being present at work during a pandemic is associated with employees’ perception of risks related to personal and work activities [44]. Implementation of precautionary behaviour at work based on accurate workplace safety perceptions of employees can supplement other organisational measures of risk prevention. Knowing employees’ perception of organisational non-technical skills is an essential resource and could be a good predictor of a return to work.

Overall, the results highlight the psychometric solidity of SAPH@W and its capacity to discriminate the positions of groups of workers along the five dimensions that underlie the perception of safety.

In particular, as well as the already known and expected gender differences, it is important to emphasise that the scale is able to measure the differences of perception based upon essential variable criteria, such as the risk level and the working condition. Those differences, as well as qualifying the utility of the scale, offer significant information for the organisation of work.

Overall, the study validated a short tool for assessing the safety conditions perceived by workers during the COVID-19 pandemic, with a specific focus on the organisational use of non-technical skills. Understanding how employees perceive risk is fundamental for managerial staff, raising awareness of the potential discrepancies between different organisational domains and allowing practices to be defined to overcome these discrepancies and potential conflicts.

While we await our final exit from the health emergency due to the ongoing vaccination campaign, the SAPH@W may be useful (for occupational health scholars, managers, practitioners) for managing the complex transition to the so-called “new normal”. It is a valid and reliable tool that allows organisations to monitor over time, e.g., after the adoption of new safety measures or training, or in different phases or pandemic waves, how and if the risk of contagion and safety perception changes, and to develop solutions that include different levels of intervention, from individual or group support [45] to interventions aimed at consolidating the feeling of safety in the workplace, even connecting to the workers’ non-technical skills.

## 5. Limitations

The study has some limitations. Firstly, the cross-sectional research design precluded the ability to investigate other relevant psychometric properties of the SAPH@W, such as its longitudinal invariance, which is a prerequisite for the assessment of change in a construct over time [36]. Furthermore, in this study, we thoroughly investigated the factor structure and the measurement invariance of the SAPH@W, but its validity needs to be further explored. Hence, future research could examine the association between the dimensions of this scale and other constructs in their nomological network. For example, we would expect a positive association between the dimensions of the SAPH@W and other positive characteristics of the work environment (i.e., job resources) [46] in terms, for example, of social support, job autonomy, and performance feedback. Similarly, based on the Self-Determination Theory [47], we would also expect a positive association between the dimensions of the SAPH@W and work-related outcomes such as job satisfaction, work engagement, and job performance. Finally, although we investigated its dimensions exclusively at the individual level in this study, it is possible that the perception of safety at work (i.e., communication, decision-making, situational awareness, and fatigue management) may be affected by processes within the organisation or at team level. Hence, future multilevel research could examine the association between safety leadership or safety climate at group level, on the one hand [48], and individual perceptions of safety at work, on the other.

## 6. Conclusions

How an organisation responds to the COVID-19 pandemic can have profound effects on the safety, health, and well-being of its employees [49].

The aim of the proposal was to bridge a gap in the literature in the field of occupational safety assessment and to provide organisations with a short tool for assessing safety conditions perceived by workers in the second and in subsequent phases of the COVID-19 pandemic. The privileged focus of our study was the subjective perception of workers on their workplace safety. The usual tools for assessing the perception of bio-risk hazards mainly focus on data of technical nature and generally concern healthcare workers. The added value of our study is the proposal of a brief questionnaire that monitors workers’ experiences with respect to the COVID-19 risk in all organisational contexts.

Moreover, in addition to other tools aimed at evaluating stress and burnout levels due, for example, to work/family conflict or social distancing [7], for workers alternating remote working and work in person, the SAPH@W can help in assessing the perceived costs and benefits in the two conditions and planning the way employees will return to work to suit the different circumstances. 

## Figures and Tables

**Figure 1 ijerph-18-05986-f001:**
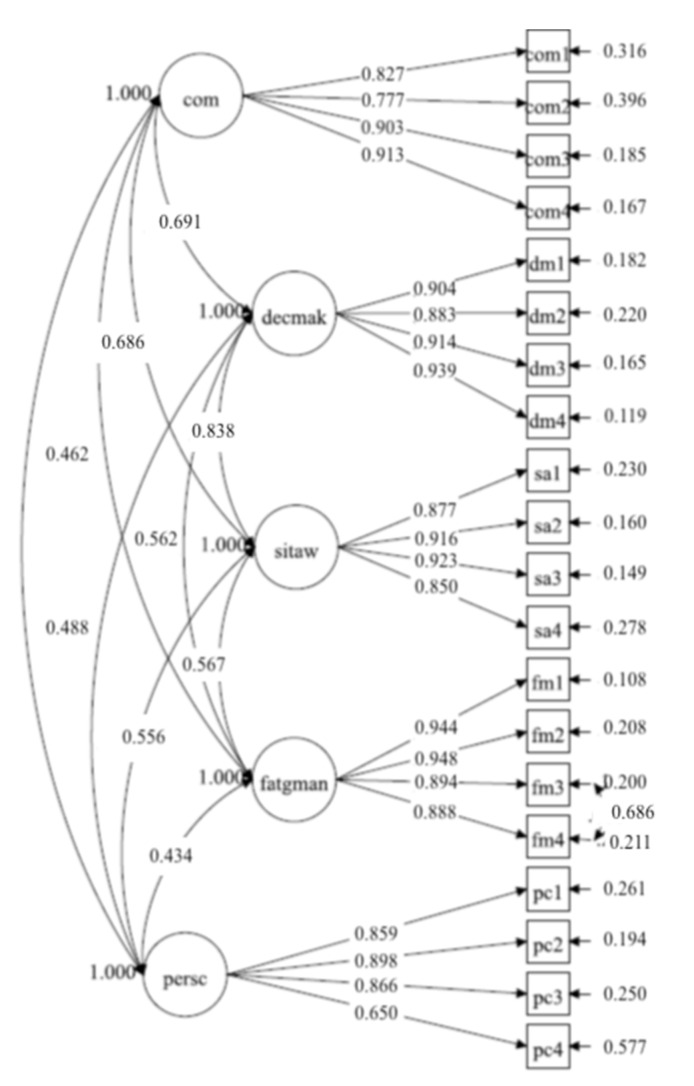
SAPH@W: CFA model on Sample-A (*n* = 542).

**Table 1 ijerph-18-05986-t001:** The SAPH@W scale.

Dimension and Item Label	Question and Item Phrasing
*Communication*	*In your opinion, in your workplace there is the opportunity to:*
COM1	Communicate effectively with the supervisor on risks related to COVID-19
COM2	Communicate effectively with colleagues on risks related to COVID-19
COM3	Have information/feedback on risks related to COVID-19
COM4	Ask for information/feedback on risks related to COVID-19
*Decision-Making*	*Regarding contagion risks by COVID-19, in your opinion, your employer organization is able to:*
DM1	Prioritize when decisions need to be made
DM2	Make quick decisions
DM3	Predict the effect of the decisions
DM4	Manage priorities
*Situational awareness*	*In your opinion, your employer organization is able to:*
SA1	Identify specific contagion risks by COVID-19 in your job
SA2	Pay attention even to details to prevent contagion risks by COVID-19
SA3	Monitor the situation to prevent contagion risks by COVID-19
SA4	Predict the future evolution of contagion hazard by COVID-19
*Fatigue Management*	*In order to contain contagion risks by COVID-19, it is important that each employee adopts specific behaviors (i.e., using PPE, keeping interpersonal distance, practicing remote working). In your opinion, your employer organization is able to:*
FM1	Recognize the possible effects of such behaviors on physical fatigue
FM2	Recognize the possible effects of such behaviors on mental fatigue
FM3	Adopt measures to reduce physical fatigue due to such behaviors
FM4	Adopt measures to reduce mental fatigue due to such behaviors
*Personal Contribution*	*Think about your feelings regarding your job in this phase. Do You feel able to:*
PC1	Provide information to other employees regarding how to tackle contagion risks by COVID-19.
PC2	Make quick decisions in front of contagion risks by COVID-19.
PC3	Understand the situation regarding contagion risks by COVID-19.
PC4	Manage mental and physical fatigue related to specific behaviours to reduce contagion risks by COVID-19

**Table 2 ijerph-18-05986-t002:** Description of samples using dichotomous variables.

	Total Sample	Sample-A	Sample-B
Sample size	1085	542	543
Age	44.2 (SD = 12.8)	42.9 (SD = 12.8)	45.5 (SD = 12.8)
Female	48.3%	48.7%	47.9%
Married	60.5%	60.0%	60.9%
Parents	50.6%	50.5%	50.6%
Risk level = high	45.9%	46.7%	45.1%
Working in presence	49.9%	52.1%	47.6%
Interacting with users/consumers	67.3%	67.7%	66.8%

**Table 3 ijerph-18-05986-t003:** SAPH@W scale: item description and scale reliability (Cronbach alpha) by sample.

	Sample-A							Sample-B						
	M	SD	S	K	Min	Max	alpha	M	SD	S	K	Min	Max	alpha
*Communication*							*0.92*							*0.90*
COM1	3.63	1.22	−0.57	−0.64	1	5		3.55	1.20	−0.48	−0.66	1	5	
COM2	3.87	1.05	−0.79	0.11	1	5		3.84	1.04	−0.84	0.31	1	5	
COM3	3.58	1.11	−0.44	−0.51	1	5		3.55	1.11	−0.48	−0.39	1	5	
COM4	3.57	1.14	−0.48	−0.46	1	5		3.45	1.15	−0.35	−0.71	1	5	
*Decision Making*							*0.95*							*0.94*
DM1	3.55	1.09	−0.46	−0.37	1	5		3.40	1.13	−0.32	−0.63	1	5	
DM2	3.48	1.17	−0.47	−0.67	1	5		3.31	1.21	−0.03	−0.80	1	5	
DM3	3.26	1.12	−0.26	−0.62	1	5		3.12	1.15	−0.22	−0.70	1	5	
DM4	3.45	1.13	−0.42	−0.53	1	5		3.33	1.15	−0.36	−0.62	1	5	
*Situational awareness*							*0.94*							*0.93*
SA1	3.51	1.11	−0.48	−0.47	1	5		3.33	1.15	−0.36	−0.62	1	5	
SA2	3.36	1.19	−0.39	−0.71	1	5		3.17	1.18	−0.19	−0.79	1	5	
SA3	3.36	1.16	−0.30	−0.78	1	5		3.22	1.19	−0.20	−0.84	1	5	
SA4	3.12	1.22	−0.10	−0.90	1	5		2.94	1.20	−0.01	−0.89	1	5	
*Fatigue management*							*0.96*							*0.97*
FM1	2.84	1.20	0.13	−0.74	1	5		2.69	1.16	0.26	−0.56	1	5	
FM2	2.82	1.17	0.09	−0.76	1	5		2.72	1.18	0.14	−0.75	1	5	
FM3	2.77	1.19	0.18	−0.82	1	5		2.62	1.18	0.27	−0.72	1	5	
FM4	2.72	1.22	0.20	−0.85	1	5		2.62	1.20	0.30	−0.76	1	5	
*Personal contribution*							*0.89*							*0.88*
PC1	3.48	1.03	−0.40	−0.23	1	5		3.52	0.92	−0.28	−0.20	1	5	
PC2	3.45	1.04	−0.47	−0.25	1	5		3.54	0.90	−0.24	−0.04	1	5	
PC3	3.55	0.98	−0.44	−0.24	1	5		3.56	0.90	−0.21	−0.23	1	5	
PC4	3.41	0.99	−0.29	−0.30	1	5		3.32	0.98	−0.08	−0.34	1	5	

Note: S is Asymmetry, K is *Kurtosis*. The standard errors for the S and K indices are 0.11 and 0.21 in Sample-A, 0.10 and 0.21 in Sample-B.

**Table 4 ijerph-18-05986-t004:** SAPH@W CFA model on Sample-B (*n* = 543): factor loadings and AVE.

Dimension	Item	Loadings	AVE
Communication	COM1	0.76	0.70
	COM2	0.68	
	COM3	0.93	
	COM4	0.93	
Decision Making	DM1	0.89	0.80
	DM2	0.88	
	DM3	0.88	
	DM4	0.93	
Situational Awareness	SA1	0.85	0.76
	SA2	0.91	
	SA3	0.89	
	SA4	0.84	
Fatigue Management	FM1	0.92	0.86
	FM2	0.95	
	FM3	0.91	
	FM4	0.92	
Personal Contribution	PC1	0.80	0.64
	PC2	0.86	
	PC3	0.84	
	PC4	0.71	

**Table 5 ijerph-18-05986-t005:** SAPH@W invariance across gender: model fit.

	df	χ^2^	rmsea	rmsea 90% ci	cfi	srmr	Δχ^2^	*p*-Value	Δcfi
Women	159	235.73	0.043	[0.032, 0.053]	0.979	0.047	-	-	-
Man	159	289.01	0.054	[0.045, 0.062]	0.967	0.035	-	-	-
Configural invariance	318	526.53	0.049	[0.042, 0.056]	0.973	0.041	-	-	-
Metric invariance	333	546.07	0.049	[0.042, 0.055]	0.972	0.042	17.80	0.27	0.001
Scalar invariance	348	567.35	0.048	[0.042, 0.055]	0.971	0.042	20.53	0.15	0.001

**Table 6 ijerph-18-05986-t006:** SAPH@W invariance across risk zone: model fit.

	df	χ^2^	rmsea	rmsea 90% ci	cfi	srmr	Δχ^2^	*p*-Value	Δcfi
Low-risk zone	159	288.79	0.052	[0.044, 0.061]	0.971	0.034	-	-	-
High-risk zone	159	224.09	0.041	[0.029, 0.052]	0.979	0.049	-	-	-
Configural invariance	318	514.10	0.048	[0.041, 0.054]	0.974	0.041	-	-	-
Metric invariance	333	541.25	0.048	[0.041, 0.055]	0.973	0.045	27.99	0.02	0.0016
Scalar invariance	348	568.77	0.048	[0.042, 0.055]	0.971	0.045	28.29	0.02	0.0016

**Table 7 ijerph-18-05986-t007:** SAPH@W invariance for working condition: model fit.

	Df	χ^2^	rmsea	rmsea 90% ci	cfi	srmr	Δχ^2^	*p*-Value	Δcfi
M1. In-presence working	159	227.79	0.041	[0.030, 0.051]	0.982	0.043	-	-	-
M2. Remote working	159	279.10	0.052	[0.042, 0.061]	0.968	0.041	-	-	-
M3. Configural invariance	318	504.19	0.046	[0.039, 0.053]	0.976	0.042	-	-	-
M4. Metric invariance(vs. M3)	333	536.84	0.047	[0.041, 0.054]	0.973	0.047	35.35	0.002	0.0023
M5. Partial metric invariance (vs. M3)	332	529.65	0.047	[0.040, 0.053]	0.974	0.046	26.41	0.02	0.0015
M6. Scalar invariance(vs. M5)	346	568.45	0.049	[0.042, 0.055]	0.971	0.047	42.46	<0.001	0.0032
M7. Partial scalar invariance (vs. M5)	345	556.58	0.048	[0.041, 0.054]	0.972	0.047	28.30	<0.01	0.0018

## Data Availability

The data presented in this study are available on request from the corresponding author. The data are not publicly available due to the Italian law on privacy.

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
