# Peer review of "Working during a Pandemic between the Risk of Being Infected and/or the Risks Related to Social Distancing: First Validation of the SAPH@W Questionnaire"

_ijerph, 2021, doi:10.3390/ijerph18115986_

Round 1

Reviewer 1 Report

Dear author / authors

I congratulate you for a good job. It seems to me a very necessary and timely work in the current context in which we find ourselves.
I will make a few simple observations that can help improve your article.
At the beginning of the manuscript you comment on the scarce bibliography that exists in this regard. Furthermore, its bibliography is conspicuously sparse. There are scientific articles even carried out in the first wave of the pandemic that address work fatigue, working conditions, work dynamics and even consequences of the absence of PPE. I think you include these articles in your research, some of them also studies conducted in Italy. I put below some references, but please, not those considered unique but perhaps as a starting point:

- Martínez-López, J.Á.; Lázaro-Pérez, C .; Gómez-Galán, J .; Fernández-Martínez, M.d.M. Psychological impact of the COVID-19 emergency on health professionals: incidence of burnout in the most critical period in Spain. J. Clin. Medicine. 2020, 9, 3029. https://doi.org/10.3390/jcm9093029
- Magnavita, N .; Chirico, F .; Garbarino, S .; Bragazzi, N.L .; Santacroce, E .; Zaffina, S. Outbreaks of SARS / MERS / SARS-CoV-2 and burnout syndrome among healthcare workers. A systematic review of umbrellas. International Journal of Environmental Research and Public Health 2021, 18, 4361
- Lázaro-Pérez, C .; Martínez-López, J.Á.; Gómez-Galán, J .; López-Meneses, E. Anxiety about the risk of death of their patients in health professionals in Spain: analysis at the peak of the COVID-19 pandemic. In t. J. Environ. Res. Public Health 2020, 17, 5938. https://doi.org/10.3390/ijerph17165938
- Franklin, P .; Gkiouleka, A. A scoping review of psychosocial risks for healthcare workers during the Covid-19 pandemic. International Journal of Environmental Research and Public Health 2021, 18, 2453
- Trumello, C .; Bramanti, S.M .; Ballarotto, G .; Candelori, C .; Cerniglia, L .; Cimino, S .; Crudele, M .; Lombardi, L .; Pignataro, S .; Viceconti, M.L .; Babore, A. Psychological adjustment of healthcare workers in Italy during the COVID-19 pandemic: differences in stress, anxiety, depression, burnout, secondary trauma, and compassionate satisfaction between frontline and non-frontline professionals. In t. J. Environ. Res. Public Health 2020, 17, 8358. https://doi.org/10.3390/ijerph17228358

Honestly, one of the weak points of their work is the lack of references from previous studies that support the need to validate the instrument they present. On the other hand, I have never seen an article published with such reduced bibliographic references in this type of scientific journal.

The methodology, like the rest of the article, is very well explained, but despite the explanation in point 2.1, was the sample homogeneous in the whole of Italian territories? What were the professional categories used? This is very valuable information that is omitted at this point. The explanation “The participants were recruited using a snowball procedure”, I don't think is enough.

I have nothing more to add, improve the context, present references to research and better explain point 2.1 of the methodology.

Author Response

RESPONSE TO REVIEWES

REV1.

At the beginning of the manuscript you comment on the scarce bibliography that exists in this regard. Furthermore, its bibliography is conspicuously sparse. There are scientific articles even carried out in the first wave of the pandemic that address work fatigue, working conditions, work dynamics and even consequences of the absence of PPE. I think you include these articles in your research, some of them also studies conducted in Italy. I put below some references, but please, not those considered unique but perhaps as a starting point:

Thank you. we know that there are many recent articles on health professional, but our study is focused on people employed in different sectors (all sectors with the exception of heath care). To our knowledge, the specific topic we faced, i.e. the perception of safety at work during the pandemic among workers not employed in the health care sector, still remain unexplored. We added the reference you suggested.

- Martínez-López, J.Á.; Lázaro-Pérez, C .; Gómez-Galán, J .; Fernández-Martínez, M.d.M. Psychological impact of the COVID-19 emergency on health professionals: incidence of burnout in the most critical period in Spain. J. Clin. Medicine. 2020, 9, 3029. https://doi.org/10.3390/jcm9093029
- Magnavita, N .; Chirico, F .; Garbarino, S .; Bragazzi, N.L .; Santacroce, E .; Zaffina, S. Outbreaks of SARS / MERS / SARS-CoV-2 and burnout syndrome among healthcare workers. A systematic review of umbrellas. International Journal of Environmental Research and Public Health 2021, 18, 4361
- Lázaro-Pérez, C .; Martínez-López, J.Á.; Gómez-Galán, J .; López-Meneses, E. Anxiety about the risk of death of their patients in health professionals in Spain: analysis at the peak of the COVID-19 pandemic. In t. J. Environ. Res. Public Health 2020, 17, 5938. https://doi.org/10.3390/ijerph17165938
- Franklin, P .; Gkiouleka, A. A scoping review of psychosocial risks for healthcare workers during the Covid-19 pandemic. International Journal of Environmental Research and Public Health 2021, 18, 2453
- Trumello, C .; Bramanti, S.M .; Ballarotto, G .; Candelori, C .; Cerniglia, L .; Cimino, S .; Crudele, M .; Lombardi, L .; Pignataro, S .; Viceconti, M.L .; Babore, A. Psychological adjustment of healthcare workers in Italy during the COVID-19 pandemic: differences in stress, anxiety, depression, burnout, secondary trauma, and compassionate satisfaction between frontline and non-frontline professionals. In t. J. Environ. Res. Public Health 2020, 17, 8358. https://doi.org/10.3390/ijerph17228358

Honestly, one of the weak points of their work is the lack of references from previous studies that support the need to validate the instrument they present. On the other hand, I have never seen an article published with such reduced bibliographic references in this type of scientific journal.

The methodology, like the rest of the article, is very well explained, but despite the explanation in point 2.1, was the sample homogeneous in the whole of Italian territories? What were the professional categories used? This is very valuable information that is omitted at this point. The explanation “The participants were recruited using a snowball procedure”, I don't think is enough.

WE BETTER EXPLAINED SAMPLE COMPOSITION SPECIFYING PARTICIPANTS PROVENIENCE (geographical areas) AND OCCUPATION.

Reviewer 2 Report

The authors evaluated working during pandemic between the risk of being infected and/or the risks related to social distancing using the SAPH@W questionnaire. This paper is well designed and interesting results of SAPH@W regarding with gender, ecological risk level, and type of occupation. This study must be valuable for COVID-19 psychometric risk in organizational situations.

The study is overall well-written but we have some specific comments.

Abstract:

Line 7: I am not sure of the meaning of (3).

Limitation and Implications:

This section should be only limitation. The contents of implications might be moved to Discussion section.

Author Response

RESPONSE TO REVIEWES

REV2.

Abstract:

Line 7: I am not sure of the meaning of (3). Thank you. We have clarified the sentence you have noticed and improved the abstract.

Limitation and Implications:

This section should be only limitation. The contents of implications might be moved to Discussion section. We modified the text as suggested.

Kind regards

Barbara

Reviewer 3 Report

This study target important aspects of organizational situations for workers during the COVID-19 pandemic. To improve the safety of working environments against infection, evaluation of SAPH@W from this study, which includes non-technical skills, would be valuable. The manuscript is well written. The followings are minor points with which the manuscript would be easier to understand for readers if addressed, I believe.

Introduction:

Lines 91-94: the SAPH@W suddenly appeared without explanation. I felt that it is better to be explained as follows, for example:

To bridge this gap, we propose SAPH@W, which is …(with a short explanation), to assess workers’ perceptions of safety, also referring to NTS domains. The objective of this study is to validate the new tool, SAPH@W using CFA.

Materials and methods:

Line 109, “Rt”: should be spelt out and explained such as “effective reproduction number, an indicator of the disease spread level”.

Also, please clarify the definitions of high- and low-risk zones.

Lines 112-116: please consider adding “using Confirmatory factor analysis (CFA)” and “multiple-group CFA” as follows because the objectives here link to the statistical methodology.

the first sample was used to test the item reliability and factorial structure of the SAPH@W using Confirmatory factor analysis (CFA), while the second sample was used to determine its measurement invariance across groups defined by gender, ecological risk level (low-high) and type of occupation (in person vs. remote working - full or partial -) using multiple-group CFA.

Lines 155-164: please consider summarising these 1-5 domains with 20 items (not as an appendix) as a table since they are the main contents evaluated and important in this study.

Line 170, “SAPH@”: typo

Results:

Line 205: is “the average of the response scale” 2.5? If so, please consider rewrite such as “the average of the response scale (i.e., 2.5)”

Lines 209-210, “the K values of all items are < 0”: but the K value for item COM2 is 0.11 (> 0).

Line 211-213, Table2: please add notes for Cronbach alpha. (In the PDF that I am reviewing, there are pictographs for the Cronbach alpha. Are they misconversion?)

Line237, Figure 1: please add explanations for the abirritations (com, decmak, sitaw, etc.)

Line 247: the term “Model 2” suddenly appeared. Please add an explanation.

Line 257, Table 3: please consider adding explanations for items (COM, DM, etc.)

Lines 273-275, lines 296-297, and section 3.4.3: please consider making a result table for the latent means with statistically significant levels for the final models in multi-group CFAs as they are important interpretation in this study.

Lines 299-321, section 3.4.3: No result for the latent mean, although discussed in the discussion as long as I understand. Is this because they are partial measurement invariance?

In (http://essedunet.nsd.uib.no/cms/topics/immigration/2/all.html), it is described that if at least two items per latent variable are equivalent, latent mean comparisons can be validly made across groups. If this reference is acceptable, please consider adding this information where appropriate.

Author Response

REV3.

Introduction:

Lines 91-94: the SAPH@W suddenly appeared without explanation. I felt that it is better to be explained as follows, for example:

To bridge this gap, we propose SAPH@W, which is …(with a short explanation), to assess workers’ perceptions of safety, also referring to NTS domains. The objective of this study is to validate the new tool, SAPH@W using CFA.

-->We modified the text following your indication.

Materials and methods:

Line 109, “Rt”: should be spelt out and explained such as “effective reproduction number, an indicator of the disease spread level”.

-->Thank you, we inserted the specification you suggested.

Also, please clarify the definitions of high- and low-risk zones.

-->We inserted the definition.

 Lines 112-116: please consider adding “using Confirmatory factor analysis (CFA)” and “multiple-group CFA” as follows because the objectives here link to the statistical methodology.

-->DONE

the first sample was used to test the item reliability and factorial structure of the SAPH@W using Confirmatory factor analysis (CFA), while the second sample was used to determine its measurement invariance across groups defined by gender, ecological risk level (low-high) and type of occupation (in person vs. remote working - full or partial -) using multiple-group CFA

-->DONE

 Lines 155-164: please consider summarising these 1-5 domains with 20 items (not as an appendix) as a table since they are the main contents evaluated and important in this study. 

-->DONE

Line 170, “SAPH@”: typo

-->CORRECTED

 Results:

Line 205: is “the average of the response scale” 2.5? If so, please consider rewrite such as “the average of the response scale (i.e., 2.5)”

-->DONE

 Lines 209-210, “the K values of all items are < 0”: but the K value for item COM2 is 0.11 (> 0). 

-->CORRECTED

 Line 211-213, Table2: please add notes for Cronbach alpha. (In the PDF that I am reviewing, there are pictographs for the Cronbach alpha. Are they misconversion?) 

-->CORRECTED

 Line237, Figure 1: please add explanations for the abirritations (com, decmak, sitaw, etc.)

-->DONE

 Line 247: the term “Model 2” suddenly appeared. Please add an explanation. ----->CORRECTED

Line 257, Table 3: please consider adding explanations for items (COM, DM, etc.) -->Now the scale is  presented in table 1 and so the item labels are self evident.

Lines 273-275, lines 296-297, and section 3.4.3: please consider making a result table for the latent means with statistically significant levels for the final models in multi-group CFAs as they are important interpretation in this study.

--> We inserted some dedicated sentences in the main text and added a table in the supplementary material.

Lines 299-321, section 3.4.3: No result for the latent mean, although discussed in the discussion as long as I understand. Is this because they are partial measurement invariance?

-->WE inserted the results.

In (http://essedunet.nsd.uib.no/cms/topics/immigration/2/all.html), it is described that if at least two items per latent variable are equivalent, latent mean comparisons can be validly made across groups. If this reference is acceptable, please consider adding this information where appropriate

We have really appreciated your useful observations. You have contributed to improve the article. Thank you so much.

Kind regards

Barbara